# Predictive Analysis of Maxillary Canine Impaction through *Sella Turcica* Bridging, *Ponticulus Posticus* Calcification, and Lateral Incisor Anomalies: A Retrospective Observational Study

**DOI:** 10.3390/mps5060091

**Published:** 2022-11-21

**Authors:** Rosanna Guarnieri, Francesca Germanò, Federica Altieri, Michele Cassetta, Camilla Grenga, Gabriella Padalino, Roberto Di Giorgio, Ersilia Barbato

**Affiliations:** Department of Oral and Maxillofacial Sciences, School of Dentistry, Sapienza University of Rome, 00161 Rome, Italy

**Keywords:** impacted tooth, *ponticulus posticus*, *sella turcica*, dental radiography, interceptive orthodontics

## Abstract

Maxillary canine impaction is an increasing dental anomaly and is often related to other dento-skeletal anomalies. The aim of this work is to support the clinician in evaluating the relationship between a displaced maxillary canine and clinical (the features of lateral incisors)/skeletal (*ponticulus posticus* and *sella turcica* bridging) anomalies through orthopanoramic radiographs, lateral cephalograms, and plaster casts to identify the parameters that best predict maxillary canine impaction. A retrospective observational study was carried out on the analysis of the medical records, radiographic findings (panoramic radiographs and lateral cephalograms), and plaster casts of 203 orthodontic patients divided into a case group, with at least one impacted maxillary canine, and a control group, without an impaction. A chi-square test and logistic regression analysis were used to analyze the data. A statistically significant association was found between the impaction of the maxillary canine and the female sex, the bridging of the *sella turcica*, the *ponticulus posticus* calcification, and the anomaly of the lateral incisor; a logistic regression revealed that these significant variables were found to be positive predictors of impacted maxillary canines, particularly in reference to the impaction in the palatal area. Finding one of these clinical and radiographic elements can represent a predictive sign of the possible impaction of the maxillary canine.

## 1. Introduction

Maxillary canine impaction incidence has significantly increased in recent years and has become an extensively treated topic in the literature. Considering some of the most significant articles about this topic published, it was found that the incidence of the maxillary canine’s impaction ranges from 0.97% to 7.1% [1,2,3,4,5,6,7,8,9]. This increase in incidence is likely due both to the improvement of the socio-economic level of the population, subjected to more frequent dental examinations, and to the development of radiological methods, which allow the acquisition of radiograms in a simple and cheap way, with low absorption of radiation [10].

The definition of an “impacted tooth”, proposed by Thilander B. and Jakobsson S.O., is “a tooth whose eruption is considerably delayed and for which there is clinical or radiographic evidence that further eruption may not take place” [2]. Regarding maxillary canine impactions, they occur 10 or 20 times more than mandibular impaction, females have a higher prevalence than males, with a variable ratio between 2:1 and 3:1, and a unilateral impaction is more common than a bilateral, with a ratio of 5:1 [11,12]. Peck et al. reported a ratio of 3:1 of palatal maxillary impactions compared to buccal; Guarnieri et al. reported that 72% of impactions occur in the palatal area, followed by 19% in the buccal area and 9% in the midalveolus [8,13].

Different theories have been proposed on the etiology of impacted maxillary canines. The lack of space in the maxillary arch is frequently associated with impaction in the buccal area [14]. In addition, there are two widely debated theories about palatal impaction: the guidance theory and the genetic theory. Regarding the first one, numerous authors have suggested that, with the agenesic, conoid, or microdontic lateral incisor, the permanent canine fails to erupt because it loses the normal guidance guaranteed by the distal part of the root of the lateral incisor [15]. However, this theory does not explain palatally impacted canines adjacent to normally developed incisors [9]. Canine anomalies are regarded as complex traits where the transmission of the trait is probably due to an autosomal dominant segregation model, with incomplete penetrance and variable expressivity [16]. In genetic theory, the palatal dislocation of the canine is frequently associated with other genetically determined dental anomalies, such as hypoplasia, dimension anomalies, and/or agenesis of the maxillary lateral incisors, aplasia of other teeth, and first molar infraocclusion [7,17,18,19,20]. According to a recent study in the literature, the prevalence of dental anomalies is around 20.9%, and the most frequent anomalies are the displacement of the maxillary canine (7.5%) [21]. Therefore, an early diagnosis of any of these dental anomalies could also indicate an increased risk for the future appearance of others and represents an important early clinical sign of a potential impaction of palatal canines [19].

The presence of easily identifiable skeletal anomalies in the lateral cephalogram, such as *sella turcica* bridging and *ponticulus posticus*, can also be associated with dental anomalies, especially palatal canine impaction [22,23].

It has been found that impacted maxillary canines and these skeletal anomalies share the same embryological origin: neural crest cells [24].

The failure to prevent maxillary canine impaction requires both orthodontic and surgical treatment, which can currently be through a minimally invasive approach, ultrasonic surgery, and through using a CAD/CAM surgical approach [25,26,27].

Impaction of the maxillary canine requires an early diagnosis for an effective interceptive intervention, planned through a careful evaluation of all the anamnestic, clinical, and radiographic elements, to increase the success rates of the spontaneous eruption of the permanent canine.

The aim of this work is to evaluate the relationship between displaced maxillary canine patients and clinical (the features of lateral incisors)/skeletal (*ponticulus posticus* and *sella turcica* bridging) anomalies through orthopanoramic radiographs, lateral cephalograms, and plaster casts to identify the parameters that best predict maxillary canine impaction.

The null hypothesis is that there is no increased prevalence of clinical and skeletal anomalies in patients with impacted maxillary canines.

## 2. Materials and Methods

### 2.1. Study Design

A retrospective observational study was carried out through the analysis of the medical records, radiographic findings (both panoramic radiographs and lateral cephalograms), and plaster casts of orthodontic patients treated in the Department of Orthodontics of the Sapienza University of Rome.

This study was performed following the approval of the regional Ethical Review Board of the “Umberto I” General Hospital of Rome (Rif. 3755).

### 2.2. Sample Size Calculation

The sample size was established following a power analysis carried out with the GPower program, which showed that the minimum number of subjects to be included in the analysis was 100 (Power = 0.80; α = 0.05; Effect size = 0.30).

The subjects numbered 203 and were selected among 1674 patients screened and treated in the Department of Oral and Maxillo-Facial Sciences over a period of 24 months from July 2019 to July 2021, according to a standardized and existing protocol that provided for:-An initial visit consisting of an intraoral and extraoral evaluation of the patients;-A second visit focused on radiographic evaluation (orthopantomography and lateral cephalogram);-A third visit for orthodontic impressions to have plaster models for the case study.

### 2.3. Inclusion/Exclusion Criteria

The inclusion criteria were the presence of intact casts, x-rays of good quality, visibility of the first four cervical vertebrae, especially the atlas, and clear visibility of the *sella turcica* on the lateral cephalogram. The exclusion criteria included medical records with incomplete information, syndromic patients, systemic pathologies, severe allergies, radiotherapy or ongoing chemotherapies, head and neck surgeries, and previous orthodontic treatment.

The x-rays analyzed were the earliest ones in patients with an established diagnosis of maxillary canine impaction.

The final sample consisted of 203 subjects (94 males and 109 females) aged between 10 and 16 years and divided into a case group of 101 subjects with at least one impacted maxillary canine with no preference for depth, position, inclination, or location (32 males and 69 females with a mean age of 11.77 years). The control group was selected sequentially starting from the dates described; the same inclusion and exclusion criteria that were valid for the study group were used, except for the presence of the impacted canine. The control group consisted of 102 subjects without an impacted maxillary canine (62 males and 40 females with a mean age of 10.74 years). The diagnosis of the maxillary impacted canine was made after 11 years of age on the basis of the clinical examination and available radiographic examinations and confirmed visually during surgery in accordance with established standardized techniques. The criteria of Ericson and Kurol, in the three sectors’ version modified by Baccetti et al., were used in the phase of impaction diagnosis on the panoramic radiograph [3,28].

The data collection process involved the elaboration of an excel worksheet (Microsoft Excel for Mac, version 2011) for data categorization against the following variables:Generic and clinical data: surname and name, gender, date of birth, and dentition (early mixed, late mixed, permanent);Radiographic data: impacted canine of each hemimaxilla (present, absent), type of impaction (unilateral, bilateral), impacted canine (right, left, both), site of impaction (buccal, palatal, midalveolus), skeletal Class (I, II, III), facial divergence (normal, hypodivergent, hyperdivergent), characteristics of *sella turcica* (absence of calcification, incomplete calcification, complete calcification) and *ponticulus posticus* (absence of calcification, incomplete calcification, complete calcification);Plaster model data: characteristics of the upper lateral incisors (congenitally missing, conoid, microdontic, normal).

All panoramic radiographs and lateral cephalograms were examined in a dark room using an x-ray illuminator panel and were traced using a 0.003-inch-thick sheet of glossy paper and a 0.5-mm fine HB lead pencil. The casts, made in the dental laboratory with medium-hard white plaster, all with a squared symmetrical base, were examined with a manual orthodontic precision gauge caliper (Leone).

### 2.4. Analysis of Panoramic Radiograph

The retrospective analysis of the panoramic radiograph was made to evaluate the characteristics of maxillary canines with a confirmed impaction diagnosis of the case group and compare them with those of the control group not affected by impaction.

### 2.5. Analysis of Lateral Cephalogram

This radiographic investigation was used to determine cephalometric data measurements and skeletal anomalies, such as *sella turcica* bridging and *ponticulus posticus* calcification, and their classification.

#### 2.5.1. Skeletal Class

The skeletal class was calculated according to Steiner in the cephalometric analysis based on the angle ANB, formed by the intersection at point N (Nasion) of the straight lines passing through points A (Maxillary) and B (Mandibular).

-Class I: 2° to 4°;-Class II: > of 4°;-Class III: < of 2°.

#### 2.5.2. Divergence

Divergence was calculated according to Steiner in a cephalometric analysis based on the Sn-GoGn angle, the angle formed between the Sella–Nasion plane and the mandibular plane passing through the Gonion and Gnation points.

-Normodivergent: Sn-GoGn angle between 27°and 37°;-Hypodivergent: Sn-GoGn angle less than 27°;-Hyperdivergent: Sn-GoGn angle greater than 37°.

#### 2.5.3. Bridging of the *Sella Turcica*

Dimensions of the *sella turcica* were manually calculated to determine the extent of the calcification. The outline of the pituitary fossa was initially drawn from the apex of the back of the *sella* to the tubercle of the visible *sella;* an initial straight line was then drawn to measure the length of the *sella* between the tubercle and the apex of the back of the *sella*, which corresponds to the position of the *sella* diaphragm or interclinoidal distance; finally, a second straight line was drawn for the greater antero-posterior diameter of the *sella turcica*, identified as the distance between the tubercle of the *sella* and the most posterior point of the internal wall of the pituitary fossa. (Figure 1).

Therefore, to quantify the severity of the bridging, the standard score scale introduced by Leonardi et al. was applied, which compares the length with the greatest anterior-posterior diameter [23]:-Class I (absence of calcification): when the length is greater than or equal to three- quarters of the greatest anteroposterior diameter;-Class II (incomplete or partial calcification): when the length is less than or equal to three-quarters of the greatest anteroposterior diameter;-Class III (complete calcification): when the diaphragm of the *sella* is evident on the radiograph (Figure 2A–C).

#### 2.5.4. *Ponticulus Posticus*

The classification of the level of ossification of the atlanto-occipital ligament is built on the visual assessment of the first four cervical units of the spine (Figure 3). The extent of calcification was recorded and assessed according to a standardized score scale (Figure 4A–C) [23]:-Class I: absence of calcification;-Class II: incomplete or partial calcification;-Class III: complete calcification.

### 2.6. Analysis of Orthodontic Plaster Casts

The plaster casts were studied to measure the maxillary lateral incisors using a manual caliper, evaluating their mesio-distal major diameter. The characteristics of both lateral incisors in patients in the control group and case group were analyzed. Information regarding the characteristics of the maxillary lateral incisors and their presence or absence was reported. Given the information according to the Becker classification, the elements have been divided according to the morphology into [16]: -Absent;-Conoids, when the mesio-distal width is greater than the cervical margin;-Microdontics, when the mesio-distal width is equal to or less than its maxillary counterpart;-Normal, when the mesio-distal width is greater than that of its maxillary counterpart,

#### Statistical Analysis

A single operator (F.G.), who had at least two years of experience in evaluating radiograms, collected all the data and examined the radiographic images. They were subjected to a second random evaluation by the same operator after a period of one month to verify the reliability of the initial observation results. It was possible to calculate the error between the two measurements using the Houston method [29]. The difference between the two observations produced a statistically insignificant error, with an equality rate of 100% for the measurements of the impacted canine, 98.8% for those of the *sella turcica*, and 98.5% for the classification of the *ponticulus posticus*.

In order to improve the reliability of the performed measurements, a second operator (C.G.) reassessed the bridging of the *sella turcica*, *ponticulus posticus*, and lateral incisor anomalies. Cohen’s kappa statistic was conducted to verify the concordance of the analyzed qualitative variables. The test result demonstrated substantial intra-examiner agreement between the two observers (Kappa > 0.90); no significant errors were found between the two analyses.

Descriptive statistics were calculated, and the results from the case and the control group were compared.

A chi-square test was used to evaluate associations between the impaction of the maxillary canine and collected data.

Logistic regression analysis (the method enter Hosmer–Lemeshow test) was used to estimate the likelihood of maxillary canine impaction and analyze the variables age, gender, type of impaction (palatal/buccal), *sella turcica* bridging, *ponticulus posticus*, and the characteristics of the lateral incisors.

The significance level was set at 0.05.

All statistical analyses were performed with Statistical Package for the Social Sciences software (IBM Corp. Released 2017. IBM SPSS Statistics for Windows, Version 25.0. Armonk, NY, USA: IBM Corp.).

## 3. Results

The descriptive statistics and results of the chi-square test comparing the case group and the control group regarding gender, divergence, and skeletal class are represented in Table 1.

The general and clinical information of all subjects were analyzed using a chi-square test (*p* < 0.001) and resulted in a statistically significant distribution of the groups, with a positive association between the impaction of the maxillary canine and the female sex.

No statistically significant association emerged from the chi-square test regarding divergence (*p* = 0.233) and skeletal class (*p* = 0.343) with maxillary canine impaction.

Analyzing the intragroup characteristics of the impacted canine subjects, only the association between the impaction of the maxillary canine with the palatal site (chi-square, *p* < 0.001) emerged as significant (Table 2). No statistically significant association emerged from the chi-square test (*p* = 0.547) between canine impaction and the right, left, and bilateral typology, resulting in a homogeneous distribution across the sample (Table 3).

### 3.1. Bridging of the Sella Turcica

As can be seen from Table 2, in the case group consisting of 101 subjects, 33.7% present a Class I bridging (the absence of bridging and normal *sella turcica*), 52.5% present Class II (calcification of the partial saddle), and 13.9% present Class III (complete calcification); in the control group, 68.6% had no calcification or normal *sella turcica* (Class I), 25.5% had partial saddle calcification (Class II), and 5.9% had complete calcification (Class III). According to the chi-square test, the distribution of classes in the sample differs between the two groups in a statistically significant manner (*p* < 0.001), and the analysis of the standardized residuals indicates that the cases are classified mainly in Class II and the controls in Class I. 

In the case group, 66.3% were affected by the bridging of the *sella turcica* (Class II and Class III); in the control group, on the other hand, 31.4% were affected too. The difference between the two groups was statistically significant (*p* < 0.001).

There was no significant difference in gender distribution (*p* = 0.232).

The association between the bridging of the *sella turcica* and divergence was also evaluated, and it was found that the chi-square test was not significant (*p* = 0.265).

Regarding the skeletal class, it emerged that patients suffering from partial and complete *sella turcica* bridging (Classes II and III) were equally distributed in skeletal Classes I and II. The distribution was not significant (*p* = 0.876).

Finally, with reference to the type of impaction, the difference is significant (*p*< 0.001). The analysis of the standardized residuals reveals a greater association with palatal impaction.

### 3.2. Ponticulus Posticus

The presence of the *Ponticulus Posticus* anomaly showed a statistically significant difference between the case and control groups (*p* = 0.010) (Table 4).

In the case group, 49.5% had Class I (the absence of an anomaly), 30.7% had Class II, and 19.8% had Class III; in the control group, 66.7% did not have any vertebral anomaly (Class I), 26.5% had Class II, and 6.9% had Class III.

According to the chi-square test, the distribution of the saddle calcification classes differed between the two groups (*p* = 0.010). 

Regarding the association between *Ponticulus Posticus*, sex, divergence, and skeletal class, the results were not statistically significant (*p* = 0.259, *p* = 0.368, and *p* = 0.728)

The distribution by site, according to the chi-square test, was slightly statistically significant (*p* = 0.048).

### 3.3. Results from the Analysis of the Plaster Casts

The results of the analysis of the plaster casts are shown in Table 5. The chi-square test between the case and control groups was significant (*p* = 0.047). 

### 3.4. Logistic Regression Analysis

To perform a logistic regression analysis, the dichotomous variables of the presence of the bridging of the *sella turcica*, *ponticulus posticus*, anomaly of the lateral incisor, and female sex were selected, as their association with the impaction of the maxillary canine was statistically significant in the chi-square test. The aim was to evaluate to what degree these factors can predict the possibility that a subject can present maxillary canine impaction and that it is buccal or palatal.

Table 6 shows the positive predictive power of the variables calculated through the logistic regression where the case and control groups were the dependent variable. The independent variables were found to lead to a greater probability of a subject being part of the case group and, consequently, to predict the maxillary canine impaction.

Specifically, it was decided to analyze the buccal or palatal impaction site as a dependent variable.

In Table 7, where the dependent variable was the buccal site, none of the predictors used could evaluate the possibility that a subject would be classified in the group of buccal impactions, despite the bridging of the *sella turcica* coming close to statistical significance.

In Table 8, the dependent variable was the palatal impaction, and it was found that the independent variables that led to a prediction of this type of impaction were the female sex, lateral incisor anomaly, and the bridging of the *sella turcica*. *Ponticulus posticus* was less significant.

Finally, the Hosmer–Lemeshow test (0.592) confirmed that the model has a good fit.

## 4. Discussion

This retrospective observational study was conducted to analyze the relationships between the impacted maxillary canine and clinical and radiographic parameters and to create a predictive analysis of impacted maxillary canines. The simultaneous analysis of the *sella turcica* and of the *ponticulus posticus* was carried out on a sample of 203 subjects (101 cases and 102 control), which, at the time of writing, is the largest in the literature.

The data emerging from the descriptive analyses of the sample reveal a higher prevalence of impacted maxillary in female subjects than in males, which is in accordance with the results reported in the literature (2:1; 3:1). A palatal site was found to be more greatly associated with the impaction of the maxillary canine than a buccal one (chi-square, *p* < 0.001) and, therefore, in line with the frequency detected by Peck S. [12]. According to Mercuri E. et al., the palatal dislocation of a canine is rarely seen as an isolated symptom but is accompanied by genetically determined dental abnormalities, such as hypoplasia and/or agenesis of the maxillary lateral incisors [7]. Indeed, a statistically significant association has been found between the anomaly of the lateral incisor (congenitally missing, conoid, or microdontic) analyzed on plaster casts and the impaction of the canine (chi-square, *p* = 0.047), with a strong prevalence of microdontic elements, as reported in the work by Baccetti [19].

The results that emerged from the analysis of the lateral cephalograms reveal that the presence of the bridging of the *sella turcica* and the *ponticulus posticus* are significantly and positively correlated with the presence of the impacted maxillary canine, which is in agreement with Leonardi R., et al. [22,23].

The anomalies of the cervical vertebrae have been described in different syndromes, and, more recently, associations have been reported between cervical vertebrae anomalies and maxillary malformations [30,31]. On the other hand, a high frequency of bridging of the *sella turcica* has been described in patients with severe craniofacial deviations and dental anomalies and in other subjects with other disorders and syndromes [23,32,33].

In fact, an increased prevalence (50.5%) of complete and incomplete *ponticulus posticus* was observed in the case group compared to the prevalence (33.5%) observed in the control group; this result is consistent with the literature [34]. An increased prevalence (66.3%) of partial and complete calcification of the *sella turcica* was found in patients in the case group compared to those in the group with normally erupted maxillary canines (31.4%). These data were also consistent with the radiographic data of the *sella turcica* bridging reported in subjects with dental anomalies [22].

These results show that skeletal anomalies in cephalometric radiographs are associated not only with malformations of the maxillaries and with craniofacial syndromes but also with dental anomalies such as the impaction of the maxillary canine. This correlation, confirmed in our study by logistic regression analysis, appears to be due to the involvement of neural crest cells and/or homeobox or hox genes during the early stages of development and the formation of the *sella turcica* and of the teeth [24]. Similarly, neck and shoulder development is affected; a mutation of a homeobox gene could be responsible for a congenital anomaly of the cervical vertebrae [35]. Since the teeth, skull, and cervical part of the spine are all affected in their development by the cells of the neural crest, none of these cell disturbances should interfere with the development of one or more of these systems under their influence.

In this study, gender was not positively associated with the presence of *sella turcica* bridging; this finding is in contrast with a previous piece of work [36]. Similarly, no statistically significant association was found between *ponticulus posticus* and gender.

Instead, a positive association was found between the palatal impacted maxillary canine and the presence of *ponticulus posticus* and bridging of the *sella turcica* as partial calcifications (Class II).

According to the study by Baccetti, the association between dental anomalies and the impaction of the palatal canine can allow for early diagnosis if only one of the related anomalies is present [19]. Taking up this concept, the results of this study reveal a statistically significant correlation between an impacted maxillary canine and skeletal anomalies, such as *ponticulus posticus* and the bridging of the *sella turcica*, that could, therefore, also be considered predictive factors.

It was found, through logistic regression analysis, that the impaction of the maxillary canine in the palatal area occurs in conjunction with *sella turcica* bridging, *ponticulus posticus*, the female sex, and the presence of a morphological anomaly of the lateral incisor.

Many of the longitudinal radiographic studies on *sella turcica* morphology have found that the growth of the *sella turcica* ceases at an early age and that from the time of puberty to late teens, there are small morphological variations in the depth and diameter but not in length [37,38].

At the same time, *ponticulus posticus*, in its complete and partial calcification, has been observed in children aged approximately 5–6 years until prepubertal age (12–13), with no evidence of late degeneration [39]. These skeletal anomalies, having an early appearance, as well as the presence of anomalies of the lateral incisor (7–8 years) and sex, could be considered risk factors that allow the clinician to make an early diagnosis and, therefore, to intercept and treat the eruption disorder at an early stage.

To our knowledge, this is the first work that studies the association between skeletal and dental anomalies and considers this number of variables.

Regarding the limitations of the study, it is desirable to use an even larger sample to improve the reliability of the results. It is also important to underline that there is a potential risk of data bias due to no randomization/blinding protocols.

Finally, the null hypothesis was rejected.

It would be interesting for the future, in addition to strengthening the use of the early prognostic factors identified in this study through randomized clinical trials, to identify the etiopathogenesis through future clinical molecular studies.

## 5. Conclusions

This retrospective observational study revealed, through logistic regression analysis, that the variables statistically significantly associated with the impaction of the canine, such as the bridging of the *sella turcica*, the *ponticulus posticus*, the female sex, and the presence of a morphological anomaly of the lateral incisor, are positive predictors of maxillary canine impaction, especially in the palatal area.

In conclusion, taking into account that these clinical and radiographic elements, namely, *sella turcica* bridging, *ponticulus posticus*, lateral incisor anomalies, and the female sex, are detectable at an early age and unchangeable, finding one of these elements can represent a predictive sign of the possible impaction of the maxillary canine and prompt the clinician to adopt adequate and targeted preventive measures.

## Figures and Tables

**Figure 1 mps-05-00091-f001:**
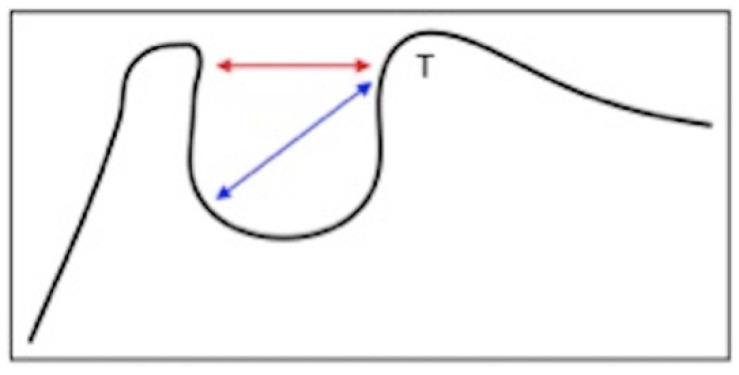
Scheme of measurements of *sella turcica* to determine the extent of the calcification. Black line: outline of the pituitary fossa. Red straight line: antero-posterior diameter of pituitary fossa as the greatest distance between the tubercle and the highest part of the back of the *sella*. Blue straight line: interclinoidal distance (length) of the *sella turcica*, drawn between the tubercle of the *sella* and the lowest point of the back wall of the pituitary fossa. T = *tuberculum sellae*.

**Figure 2 mps-05-00091-f002:**
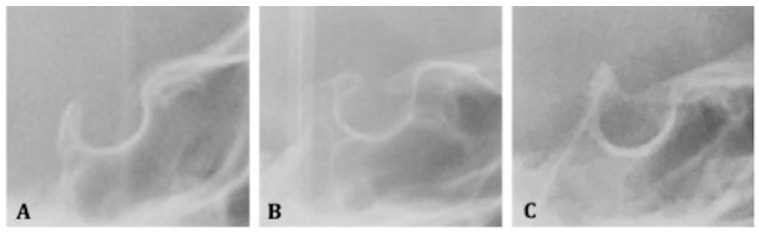
Examples of calcification of *sella turcica* to classify the bridging. (**A**) Class I of *sella turcica* bridging: absence of calcification; (**B**) Class II of *sella turcica* bridging: incomplete or partial calcification; (**C**) Class III of *sella turcica* bridging: complete calcification.

**Figure 3 mps-05-00091-f003:**
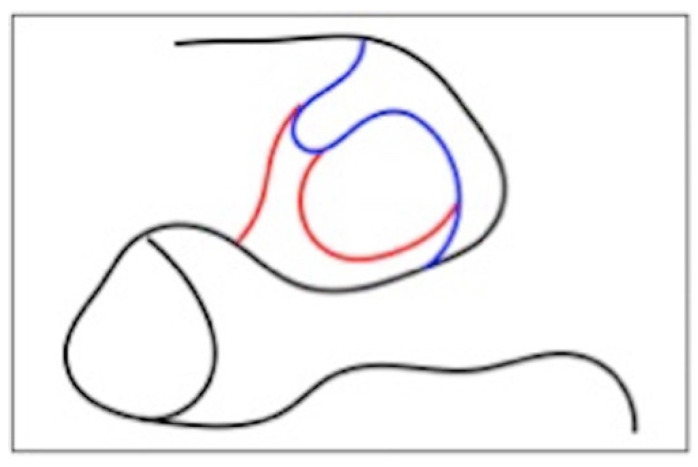
Visual scheme to determine the extent of ossification of the atlanto-occipital ligament. Black: outline of atlas, first vertebra. Blue: incomplete ossification bridge. Red: complete ossification bridge.

**Figure 4 mps-05-00091-f004:**
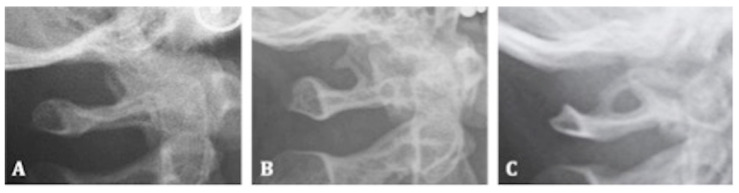
Examples of classification of *ponticulus posticus*. (**A**) Class I of *Ponticulus Posticus*: absence of calcification. (**B**) Class II of *ponticulus posticus*: incomplete or partial calcification. (**C**) Class III of pontiuclus posticus: complete calcification.

**Table 1 mps-05-00091-t001:** Descriptive statistics and results of general and clinical information comparing the case group and the control group (chi-square test, significant threshold *p* < 0.05).

	Case (101 Patients)	Control (102 Patients)	*p*-Value
**Gender**	32 (31.7%)	62 (60.8%)	<0.001
	Male	Male	(chi-square)
	69 (68.3%)	40 (39.2%)	
	Female	Female	
**Divergence**	48 (47.5%)	30 (29.4%)	0.233
	Normodivergent	Normodivergent	(chi-square)
	34 (33.7%)	42 (41.2%)	
	Hypodivergent	Hypodivergent	
	19 (18.8%)	30 (29.4%)	
	Hyperdivergent	Hyperdivergent	
**Skeletal Class**	48 (47.5%)	39 (45.5%)	0.343
	Class I	Class I	(chi-square)
	46 (45.5%)	52 (51%)	
	Class II	Class II	
	7 (6.9%)	11 (10.8%)	
	Class III	Class III	

**Table 2 mps-05-00091-t002:** Descriptive statistics and results of *Sella Turcica* Bridging (chi-square test, significant threshold *p* < 0.05).

*Sella Turcica* Bridging	Class I	Class II	Class III	*p*-Value
**Patients**	34 (33.7%)	53 (52.5%)	14 (13.9%)	<0.001
	Case	Case	Case	(chi-square)
	70 (66.6%)	26 (25.6%)	6 (5.9%)	
	Control	Control	Control	
**Gender**	54 (57.4%)	31 (33%)	9 (9.6%)	0.232
	Male	Male	Male	(chi-square)
	50 (45.9%)	48 (44%)	11 (10.1%)	
	Female	Female	Female	
**Divergence**	34 (43.6%)	35 (44.9%)	9 (11.5%)	0.265
	Normodivergent	Normodivergent	Normodivergent	(chi-square)
	40 (52.6%)	27 (35.5%)	9 (11.8%)	
	Hypodivergent	Hypodivergent	Hypodivergent	
	30 (61.2%)	17 (34.7%)	2 (4.1%)	
	Hyperdivergent	Hyperdivergent	Hyperdivergent	
**Skeletal**	42 (48.3%)	36 (41.4%)	9 (10.3%)	0.876
**Class**	Class I	Class I	Class I	(chi-square)
	53 (54.1%)	35 (35.7%)	10 (10.2%)	
	Class II	Class II	Class II	
	9 (50%)	8 (44.4%)	1 (5.6%)	
	Class III	Class III	Class III	
**Type of Impaction**	70 (32.3%)	18 (58.1%)	3 (9.7%)	<0.001
	Buccal	Buccal	Buccal	(chi-square)
	23 (33.8%)	35 (51.5%)	10 (14,7%)	
	Palatal	Palatal	Palatal	
	1 (50%)	0	1 (50%)	
	In Crest	In Crest	In Crest	

**Table 3 mps-05-00091-t003:** Intragroup descriptive statistics and results of type and site of impaction (chi-square test, significant threshold *p* < 0.05).

		*p*-Value
**Type of Impaction**	**28 (27.7%)**	**0.547**
	**Right**	**(Chi-Square)**
	38 (37.6%)	
	Left	
	35 (34.7%)	
	Bilateral	
**Site of Impaction**	31 (30.7%)	<0.001
	Buccal	(chi-square)
	68 (67.3%)	
	Palatal	
	2 (2%)	
	in crest	

**Table 4 mps-05-00091-t004:** Descriptive statistics and results of *Ponticulus Posticus* (chi-square test, significant threshold *p* < 0.05).

*Ponticulus Posticus*	Class I	Class II	Class III	*p*-Value
**Patients**	50 (49.5%)	31 (30.7%)	20 (19.8%)	0.010
	Case	Case	Case	(chi-square)
	68 (66.7%)	27 (26.5%)	7 (6.9%)	
	Control	Control	Control	
**Gender**	49 (52.1%)	30 (31.9%)	15 (16%)	0.259
	Male	Male	Male	(chi-square)
	69 (63.3%)	28 (25.7%)	12 (11%)	
	Female	Female	Female	
**Divergence**	45 (57.7%)	23 (29.5%)	10 (12.8%)	0.368
	Normodivergent	Normodivergent	Normodivergent	(chi-square)
	49 (64.5%)	20 (26.3%)	7 (9.2%)	
	Hypodivergent	Hypodivergent	Hypodivergent	
	24 (49%)	15 (30.6%)	10 (20.4%)	
	Hyperdivergent	Hyperdivergent	Hyperdivergent	
**Skeletal Class**	55 (63.2%)	21 (24.1%)	11 (12.6%)	0.728
	Class I	Class I	Class I	(chi-square)
	53 (54.1%)	32 (32.7%)	13 (13.3%)	
	Class II	Class II	Class II	
	10 (55.6%)	5 (27.8%)	3 (6.7%)	
	Class III	Class III	Class III	
**Type of Impaction**	14 (45.2%)	12 (38.7%)	5 (16.1%)	0.048
	Buccal	Buccal	Buccal	(chi-square)
	35 (51.5%)	19 (27.9%)	14 (20.6%)	
	Palatal	Palatal	Palatal	
	1 (50%)	0	1 (50%)	
	In Crest	In Crest	In Crest	

**Table 5 mps-05-00091-t005:** Descriptive statistics and results of plaster casts (chi-square test, significant threshold *p* < 0.05).

Lateral Incisors	Case Group	Control Group	*p*-Value
	202 Lateral Incisors	204 Lateral Incisors	
**Normal Shape**	140 (69.3%)	183 (89.7%)	
**Congenitally Missing**	7 (3.4%)	4 (1.9%)	
	3 unilateral	0 unilateral	
	4 bilateral	4 bilateral	
**Conoid**	18 (8.9%)	4 (1.9%)	
	4 unilateral	0 unilateral	
	14 bilateral	4 bilateral	
**Microdontic**	37 (18.3%)	13 (6.3%)	
	3 unilateral	1 unilateral	
	34 bilateral	12 bilateral	
			0.047 (chi-square)

**Table 6 mps-05-00091-t006:** Logistic regression results (case/control).

	B	E.S.	Wald	df	Sig.	Exp (B)
**Gender, female**	−1.347	0.339	15.797	1	0.000	0.260
***Sella turcica* bridging**	−1.492	0.333	20.009	1	0.000	0.225
** *Ponticulus posticus* **	−0.976	0.344	8.063	1	0.005	0.377
**Lateral incisor anomaly**	−1.265	0.395	10.234	1	0.001	0.282

**Table 7 mps-05-00091-t007:** Results of logistic regression (buccal site’s dependent variable).

	B	E.S.	Wald	df	Sig.	Exp (B)
**Gender, female**	0.469	0.422	1.234	1	0.267	1.599
***Sella turcica* bridging**	0.809	0.424	3.645	1	0.056	2.245
** *Ponticulus posticus* **	0.626	0.408	2.349	1	0.125	1.870
**Lateral incisor anomaly**	0.530	0.429	1.525	1	0.217	1.699

**B**: co-efficient for the constant; **E.S.**: standard error around the co-efficient for the constant; **Wald** chi-square statistics; **df:** degree of freedom for Wald chi-square statistics; **Exp (B)**: exponentiation of B co-efficient.

**Table 8 mps-05-00091-t008:** Results of logistic regression (palatal site’s dependent variable).

	B	E.S.	Wald	df	Sig.	Exp (B)
**Gender, female**	0.926	0.332	7.785	1	0.005	2.524
***Sella turcica* bridging**	0.978	0.324	9.141	1	0.002	2.660
** *Ponticulus posticus* **	0.485	0.327	2.194	1	0.139	1623
**Lateral incisor anomaly**	0.700	0.353	0.919	1	0.048	2.013

**B**: Co-efficient for the constant; **E.S.**: standard error around the co-efficient for the constant; **Wald** chi-square statistics; **df:** degree of freedom for Wald chi-square statistics; **Exp (B)**: exponentiation of B co-efficient.

## Data Availability

Not applicable.

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
