# Peer review of "Predictive Analysis of Maxillary Canine Impaction through Sella Turcica Bridging, Ponticulus Posticus Calcification, and Lateral Incisor Anomalies: A Retrospective Observational Study"

_mps, 2022, doi:10.3390/mps5060091_

Round 1
Reviewer 1 Report
This article entitled " Predictive Analysis of Maxillary Canine Impaction through Sella Turcica Bridging, Ponticulus Posticus Calcification and Lateral Incisors Anomalies: A Retrospective Observational Study” intends to analyze the the possible relationship between the included maxillary canines and possible clinical and/or skeletal anomalies of the patient such as abnormal anatomy of the lateral incisors, ponticulus posticus or the sella turcica through radiographs, cephalometrics and plaster models.
Introduction
- What is the probability of the relationship between impacted canines and the dental anomalies mentioned in the introduction. Is it a noteworthy percentage? Are there other authors who consider the opposite or is this statement unanimous in the medical community?
Material and method
- Within the data collected from the patients, they specify that the skeletal class was noted. We advise specifying what was the criterion/author/measure to evaluate that skeletal class.
- Was the analysis of the radiographs and study models carried out by the same operator or by several operators?
- In the case of the size of the incisors. The contralateral tooth was taken into account to assess whether it was microdontic. Was the possible case of having both microdontic teeth taken into account? How was it considered? We understand that the tooth on the side of the impacted canine was evaluated, but we recommend specifying it. In the case of the control group, which was evaluated?
Results
- Authors are advised to review table 2, 7 and 8.
- Authors are suggested to make a note below the tables with the abbreviations used.
Discussion and conclusions
After the results obtained and the comparison with previous studies, what would be the future for this field of analysis?
According to the results, at what age do the authors consider that these predictive values ​​should be analyzed or sought?
And once detected, what would be the steps to follow as a preventive measure? Should other variables be taken into account?
Author Response
Dear Reviewer, the file with the explanations relating to the requested changes has been attached.
Thanks for your consideration.

Reviewer 2 Report
It seems this is a very interesting paper. The authors have included several parameters in their analysis. The statistical analysis seems to be adequate.
There are some issues in the methodology that I would like to discuss.
It is written in the paper on page 3 lines 129-130: "The final sample consisted of 203 subjects (94 males and 109 females) aged between 129 10 and 16 years". Also on page 3 lines 136-138: "The diagnosis of maxillary impacted canine was made after 11 years 136 of age on the basis of the clinical examination and available radiographic examinations......" How they have diagnosed the impaction of canines in 10 years old subjects that they have included in the sample?
It is not clear if one observer has evaluated all the data. It seems from what it has been written that one observer only has studied the radiographs and the rest of the material and performed the measurements/evaluations. Also, it is not clear if the intra-observer agreement has been tested. All the above are disadvantages of the study and affect the accuracy of the conclusions.
Author Response

(The authors gave the same response as above.)
